# Enhancing Wearable Gait Monitoring Systems: Identifying Optimal Kinematic Inputs in Typical Adolescents

**DOI:** 10.3390/s23198275

**Published:** 2023-10-06

**Authors:** Amanrai Singh Kahlon, Khushboo Verma, Alexander Sage, Samuel C. K. Lee, Ahad Behboodi

**Affiliations:** 1Sanford School, Hockessin, DE 19707, USA; 2Pediatric Mobility Lab, Department of Physical Therapy, University of Delaware, Newark, DE 19716, USA; vkhush@udel.edu (K.V.); slee@udel.edu (S.C.K.L.); 3Independent Researcher, Boulder, CO, USA; alexander.p.sage@gmail.com; 4Neurorehabilitation and Biomechanics Research Section, Rehabilitation Medicine Department, Clinical Center, National Institutes of Health, Bethesda, MD 20892, USA

**Keywords:** gait analysis, treadmill, walking, IMU, wearable sensors, kinematics, similarity distance, Poincare, spatiotemporal, adolescents

## Abstract

Machine learning-based gait systems facilitate the real-time control of gait assistive technologies in neurological conditions. Improving such systems needs the identification of kinematic signals from inertial measurement unit wearables (IMUs) that are robust across different walking conditions without extensive data processing. We quantify changes in two kinematic signals, acceleration and angular velocity, from IMUs worn on the frontal plane of bilateral shanks and thighs in 30 adolescents (8–18 years) on a treadmills and outdoor overground walking at three different speeds (self-selected, slow, and fast). Primary curve-based analyses included similarity analyses such as cosine, Euclidean distance, Poincare analysis, and a newly defined bilateral symmetry dissimilarity test (BSDT). Analysis indicated that superior–inferior shank acceleration (SI shank Acc) and medial–lateral shank angular velocity (ML shank AV) demonstrated no differences to the control signal in BSDT, indicating the least variability across the different walking conditions. Both SI shank Acc and ML shank AV were also robust in Poincare analysis. Secondary parameter-based similarity analyses with conventional spatiotemporal gait parameters were also performed. This normative dataset of walking reports raw signal kinematics that demonstrate the least to most variability in switching between treadmill and outdoor walking to help guide future machine learning models to assist gait in pediatric neurological conditions.

## 1. Introduction

Instrumented gait analysis of walking, a primary human locomotor action, is beneficial for healthcare [1,2,3,4], robotics [5], gait biometrics [6,7], and sports performance applications [8] and is evolving rapidly due to recent technological advances [1,2,3,4,7,8]. The quantification of gait data requires precise collection and analysis of data; these processes are inextricably linked.

Ground truth precision for gait data collection and analysis has been lab-based and is well-standardized using instrumented motion capture [9,10]. Equipment used for instrumented motion capture includes high-frame-rate infrared cameras (instrumented motion capture) [11,12], electromyography (EMG) [13], and force plates [14]. However, with transformational changes in portable and wearable motion sensors that improved their reliability and accuracy while reducing their power consumption, size, and cost, gait data collection is shifting from a cumbersome and expensive laboratory-based gold standard to real-world testing [15,16]. 

With the appropriate sensor setup, large amounts of data across patient populations, terrains, walking conditions, and gait speeds can be reliably captured; with appropriate analysis and signal processing techniques, the resulting output could be helpful for diagnostic, therapeutic, and assistive gait protocols. Such potential applications include early screening for Parkinson’s [17], gait training in cerebral palsy (CP) [9], and the development of exoskeleton assistive devices for walking [11,16,18,19].

Inertial measurement units (IMUs) include motion sensors (accelerometers, magnetometers, and gyroscopes) and rechargeable batteries that enable untethered use to stream and log gait data in and out of lab environments. Commercial IMUs such as the APDM Opal (Portland, OR, USA) and Xsense MVN (Henderson, NV, USA) are now widely available for research and clinical applications and are capable of mounting on various locations on the body. Additionally, low-cost consumer IMU versions common in smartphones and watches make gait data collection and monitoring possible in many additional environments [2,20,21]. With these wearable sensors and defined gait assessment protocols, the logical transition of gait monitoring systems from instrumented motion capture laboratories to real-world applications is happening. Current untethered monitoring of gait using IMUs includes indoor treadmill walking, indoor overground walking, outdoor walking, and real-life walking during activities of daily living [22]. In addition to walking gait, other forms of locomotion such as running and jumping are also studied using IMUs for diagnostic or treatment protocols [23]. 

Despite the inherent advantages of wearables, such as affordability, accessibility, and ease of use, their application in healthcare was tempered by both a lack of standardization in data collection and data analysis [15,23,24]. In the absence of adequate systematic studies to standardize IMU-based normal and pathological gait analysis, there needs to be a consensus over which kinematic signals are best used in the proposed IMU-based gait monitoring systems [24]. 

Gait kinematics vary substantially due to many factors, including patient populations, physical environments, and cognitive loads. Leveraging the advantages of IMUs, collecting big data helps address this variability [15,16] to design better gait monitoring and therapeutic devices [2,3,24]. Regardless of the data collection methodology, in the lab or outdoors, the extensive steps required for processing large volumes of data further limit the widespread commercial application of IMUs [23,25]. To objectively measure gait, both classical statistical testing and threshold testing of specific parameters were used to identify pathological gait; however, these parameters may fail to capture the full complexity of risk factors and clinical variables. Thus, machine learning, using the entire gait curve and waveform and not just specific parameters, was employed to more accurately analyze and efficiently identify gait cycles [26].

Gait assessment with IMUs involves separating the triaxial (anterior–posterior (AP) vs. medial–lateral (ML) vs. superior–inferior (SI)) signals, such as acceleration or angular velocity, into quantitative variables [20]. In gait lab studies, medial–lateral shank angular velocity is a reliable signal for gait phase detection in pathologic gait in CP [10,27]. Furthermore, Rastegari et al. [28] looked for signals that show maximal information gain with minimal correlation (MIGMC) and noted that anterior–posterior acceleration and medial–lateral acceleration show a pairwise correlation <80% and, when used together, play an essential role in discriminating different gait in patterns machine learning algorithms. For measuring these specific acceleration signals with stand-alone IMUs (or smartphones), the real-world testing of such IMUs is necessary because there is high variability in axis orientation, making standardization of the axis orientation crucial. To deal with the orientation problem, an orientation-independent model was proposed [29]. Alternatively, in cases where multiple signals from different body locations are available, fusing some of the signals gives greater accuracy for lower limb gait analysis [30]. These latter techniques represent work that aspires to identify the ideal signal type, a defined axis, and a preferred sensor location to optimize gait monitoring models, especially for those using machine learning.

Recent progress was seen in the clinical implementation of gait monitoring systems by using machine learning, which enables greater efficiency in processing gait parameters [2,15,31,32]. Thus, current state-of-the-art post hoc gait data analysis involves data processing, data modeling, and now, machine learning algorithms [32]. Gait phase detection using machine learning, deep neural networks, and input from wearable sensors measuring acceleration has shown promise, with gait phase detection accuracy ranging from 75 to 95% in real-time [5] and nearly 99% when offline [18]. Zhen et al. used a long short-term memory–deep neural network (LSTM-DNN) algorithm with three IMUs on one leg (thigh, calf, and foot) for real-time detection. Vu et al. used an exponentially delayed fully connected neural network (ED-FNN) with one IMU (shank) for offline detection. Via similar machine learning investigations, effective algorithms can be developed to control power active assistive devices and better address patient gait pathology, including pediatric neuromuscular diseases such as cerebral palsy (CP) [2,19] and adult neuromuscular diseases such as Parkinson’s disease (PD) [17]. An advantage of these signals-based algorithms is the ability to utilize raw gait signals instead of waiting for the processed gait data, such as measuring joint angles [17], which, in turn, may allow for less processing burden, improved real-time capacity, and greater commercial adoption.

The effect of varying conditions, such as walking surfaces, age, gait speed, and load, on walking gait was studied to different degrees. For example, ample published evidence demonstrates differences between treadmill walking and indoor overground walking [33]. The gait kinematics of the pediatric population was extensively studied using the gold standard lab analysis and somewhat studied with indoor overground walking using IMUs; however, to date, the pediatric population was understudied outdoors [16,19,20,25]. Additionally, gait kinematics depends on walking speed and the age of the population [34,35,36]. To quantify speed and age-dependent changes in gait, a normative dataset of pediatric gait was created for self-selected (SS) and fast-as-possible (FAP) speeds [12,16]. Thus, when using kinematic signals for gait monitoring applications in the pediatric population, signals that remain robust and reliable at different speeds and on different walking surfaces/conditions are necessary. 

Gait metrics, such as spatiotemporal parameters [12,16,37], were statistically compared to demonstrate typical gait and deviations from typical gait. These studies, in turn, led to the introduction of standardized indices for comparing pathological and typical gaits, such as the Gait Deviation Index (GDI) [38], Gait Variable Score [39], and Movement Deviation Profile [40]. Similarity measurements were used in non-healthcare fields to compare individual subject characteristics in voice recognition, for example [41]. Recent research deployed similarity measures to quantify variability and changes in gait kinematics [42], especially to compare typical and pathological gait. In comparing gait in different conditions, image-recognition procedures [6] and similarity networks such as aggregated correlation values [28] and percentage of statistical significance [43] were studied. Standard distance-based similarity measures, such as cosine and Euclidean distance, as opposed to parameter-based measures, such as aforementioned spatiotemporal measures, are showing increased adoption [44]. Distance-based similarity measures, which consider the entire dataset and not just a few specific features, are moving toward an accepted future standard [14]. Using the conventional distance methodology, one can precisely measure similarity. However, the statistical comparison of the changes in similarity or dissimilarity across conditions is challenging. 

As IMU-based gait studies and gait similarity evolve, there has yet to be a consensus or standardization of which kinematic signals are most reliable across conditions that can be used to develop robust gait monitoring systems. In our study, we propose analysis methods that can identify the kinematic signals showing robust stability even when walking conditions change, such as variability in gait speed or physical environment, similar to real-world walking. The objectives of this study, therefore, were to provide proof of concept that (1) reliable lower limb kinematic signals derived from IMUs, such as angular velocity and acceleration, can be identified that are independent of treadmill and outdoor overground walking conditions and (2) reliable lower limb kinematic signals can be identified that are independent of walking speed, i.e., self-selected, slow, and fast speeds. Our overarching goal was to identify optimal kinematic signals as the inputs of a reliable gait monitoring system. To quantify the effect of environmental and walking speed conditions for our study group, we compared 3-dimensional patterns of angular velocity and acceleration with IMUs during individual gait cycles. We defined a control metric (baseline) against which similarities across surfaces can be compared to quantify the significance of similarity measures. This procedure uses a Z-score as proposed by GDI to quantify the level of similarity. The ultimate goal of this study is to identify specific candidate raw signals for future real-time deep learning applications.

## 2. Materials and Methods

### 2.1. Participants

Thirty typically developing adolescents (ages 8–18 years old, 20 male/10 female) volunteered as participants in this study (see Figure 1). These participants were recruited via convenience sampling. Participants aged 18 were required to give written informed consent, and participants under 18 were required to provide both written participant assent and written parental consent. Individuals who reported orthopedic or neuromuscular conditions affecting their gait and posture or marked visual and hearing deficits were excluded from the study. All participants were required to have prior treadmill walking experience and to complete a 40 min walking test to ensure gait data collected on the treadmill was reliable and any differences between indoor treadmill walking and outdoor overground walking were not due to unfamiliarity with treadmill walking. The study protocol was approved by the Institutional Review Board of Sanford School (Hockessin, DE, USA, 29 July 2022).

### 2.2. Instrumentation

Before arrival, participants were instructed to wear comfortable, closed-toe walking shoes and attire suitable for light exercise. Before starting data collection, participants were asked to strap the IMUs on the intended locations (frontal plane of the bilateral shanks and thighs) with instruction from a researcher, ensuring that the IMUs were in the correct location and did not shift with leg movement. The specific IMU mounting configuration was chosen because it was studied for children with CP [19], and we plan future work with this research protocol using data from individuals having CP. The same researcher confirmed the patient data, consent, and IMU mounting with the correct orientation for each participant. 

Gait data were measured using the Opal IMU system from APDM Wearable Technologies (v1.0, Portland, OR, USA). As per manufacturer recommendations, a standardized IMU orientation was implemented in which the IMUs were placed in the frontal plane where z = AP axis, y = ML axis, and x = SI axis, as shown in Figure 2C.

For this specific experiment, four Opal IMUs were used for each participant: one on each shank and one on each thigh. These IMUs streamed three-dimensional angular velocity and acceleration data at a sample rate of 128 Hz. Data were collected wirelessly via APDM’s MotionStudio^TM^ software using a laptop with an APDM Access Point. The data were captured and managed nearby by a laptop-equipped researcher. Indoors, participants walked on a treadmill (Bodyguard Radisson Plus, Bodyguard Fitness, Quebec, Canada), and the same treadmill was used for all trials to minimize variation or calibration issues (Figure 2A). The researcher’s laptop was on an adjacent table to minimize the probability of losing data points. Outdoors, the participants walked overground, and the researcher’s laptop was walked with the participants (Figure 2B). The same IMU setup was used in a single continuous testing session per participant, starting with indoor treadmill walking followed by outdoor overground walking, allowing for consistent data recording for each participant, as shown in Figure 2. 

Controlling for axis and standardizing axis orientation for IMUs is necessary for gait studies, and this was addressed by either standardizing multiple IMU placement [18,19,30] and walking direction [31] or creating an orientation-independent axis [29]. Axis orientation variability was minimized proactively via researcher confirmation of each IMU placement in the frontal plane (Figure 2C), avoiding turns while walking and limiting participant ambulation activities to walking only.

### 2.3. Data Collection

Self-selected speed in overground walking is quantitatively similar to treadmill walking and was chosen over a fixed speed (for the entire participant population) to allow for individual participant variability [12,45]. To determine their individual walking speeds, participants were asked to perform a standardized 10 m walking test (10 mWT) to determine self-selected speed, fast speed (20% faster), and slow speed (20% slower) [37]. Self-selected speed is also sometimes referred to as preferred walking speed (PWS), and when determined overground, it is further termed O-PWS [21].

Starting with participants, progressing to the anatomic locations of the IMUs, and culminating with the various walking trials, the data collection process is shown in Figure 3. The order for the treadmill walking trials was self-selected speed, fast speed, and slow speed, as outlined in Figure 3, with two 3 min trials at each speed. The treadmill’s speed was adjusted by a researcher, and data collection started once the treadmill reached the participant’s appropriate pre-determined self-walking speed. This protocol helped familiarize the participants with these speeds so they could more accurately replicate the speeds in the outdoor trials. Previous research shows that the difference between treadmill and overground gait decreases after 6 min of treadmill walking, representing an adaptation in gait known as the treadmill accommodation phenomenon [45]. Thus, each treadmill trial lasted three minutes, and breaks of up to two minutes were allowed between each trial as we wanted to minimize the treadmill accommodation effect that occurs at approximately 6 min of continuous walking. 

The order for outdoor walking trials (self-selected, fast, and slow) was also kept consistent and mirrored that of the treadmill trials (see Figure 3). The same outdoor flat asphalt section was used for all 30 participants to standardize the environmental load and the overground surface. All walking tests were limited to days without extreme weather (temperature range: 55–75 °F). We collected 27 min of gait kinematics (18 min treadmill and 9 min outdoor) for each of the 30 participants. A total of 90 trials (two treadmill and one outdoor session per participant) were recorded for each speed across participants.

Outdoor trials at three different speeds of 3 min each were, on average, 250–300 m (820–984 ft) in length, and no turns or pauses were performed during each trial. Because outdoor walking speed naturally fluctuates and is more difficult to control than during treadmill walking, we measured cadence as a proxy to ensure actual changes in outdoor walking speed were similar to that occurred on the treadmill. The IMUs chosen for data collection in this study have a streaming live data transmission mode, which requires an active, powered wireless access point within the wireless transmission range of the IMUs. Thus, the equipment needed for experimentation (access point, docking station, and data collection laptop) was wired to a portable battery and supported on a custom mount on a researcher’s bicycle (Figure 2B). The researcher walked with the bicycle behind the participant to maintain proximity to the sensors, thus minimizing any streaming lag or dropped data. The researcher trailed the participant to minimize any visual or environmental load on the participant’s gait.

The three-dimensional (3D) angular velocity (AV) and acceleration (Acc) of each of the four IMUs were used for our analyses. In summary, we analyzed 24 kinematic signals for each individual: four locations (left and right shank and thigh), two sensors (AV and Acc), and three axes per sensor (X = SI, Y = ML, and Z = AP). 

### 2.4. Data Processing

Data were analyzed using Python algorithms in non-proprietary software. Primary outcome measures of the kinematic signals were distance-based similarity (cosine and Euclidean distance) and Poincare analysis, both of which are curve-based. Distance measures (cosine and Euclidean distance) metrics were used to first rank the kinematic signals across the walking conditions of gait, respectively and then to measure paired comparisons of similarity across walking conditions of gait, respectively. Additionally, five spatiotemporal features (parameter-based outcomes) were used as secondary outcome measures to statistically compare the signals’ pattern between the treadmill and the overground.

Each participant’s two treadmill 3 min walking trials at each speed were appended into a single 6 min trial. The use of 2 shorter trials at each speed was employed to mitigate the effects of the treadmill accommodation phenomenon [45] and, when appended, allows for a longer trial at each treadmill walking speed. Testing at all speeds results in 90 treadmill datasets compared to 90 outdoor datasets.

#### 2.4.1. Signal Conditioning

Recorded raw data was de-identified, converted into non-proprietary CSV format, and subjected to a Python extraction function for file metadata. The first and last five seconds of data from each file were removed to minimize possible errors from the participant speeding up/slowing down. The conditioning steps were as follows.

Upon examining the raw data from the ML Shank AV, a 4th-order low-pass Butterworth filter with a cut-off frequency of 20 Hz was applied to smooth the raw signals. Then, gait cycle initiation was identified as the time point where ML shank AV crossed the zero line from a period of negative value (negative-to-positive zero-crossing in Figure 4, red dots) [9]. Therefore, gait cycle duration was defined as the time between two consecutive negative-to-positive zero-crossings, i.e., two consecutive red dots in Figure 4. To differentiate between small fluctuations around the zero line and the actual zero-crossing at the initiation of the gait cycle, and thereby, to avoid misdetection of gait cycles, the threshold for the zero-crossing detection was set at −0.3 rad/s. A pattern monitoring algorithm was used to exclude the aberrant gait cycles. This algorithm was based on the existence of a valley, its preceding peak, and the 95% confidence interval of the mean gait cycle duration. The average gait cycle duration was calculated across all participants using the 90 collected trials. The pattern recognition algorithm removed any gait cycle that fell outside the 95% confidence interval (mean gait cycle duration +/− two std). Of the 317,000 cycles identified for all 30 participants, 97% of all cycles were retained after the 95% CI filter was applied. The subject-by-subject analysis of the 95% CI filtering demonstrated that 95% to 99% cycles of each subject were retained. The gait cycle patterns of each signal were generated by normalizing the gait cycles to their duration and averaging them across all participants and all the trails. We reported the pattern as the percentage of the gait cycle by linear interpolation into 100 data points, as shown in Figure 4. A total of 24 (4 IMU locations × 2 kinematics (acceleration, angular velocity) × 3 dimensions/axes) patterns were generated.

#### 2.4.2. Similarity Analysis

We combined the data from the left and right legs for this analysis, resulting in a total of 12 kinematics signals (2 locations × 2 sensors × 3 axes). The primary and secondary outcome measures in this study are listed in Figure 5. As part of the primary outcome measures, to analyze the similarity between the treadmill and outdoor signals (across the whole curve) averaged across all participants, we used distance-based similarity measures, i.e., cosine and Euclidean. For a comparison of similarity, the aggregated data from all subjects were ranked by kinematic signal, measuring indoor versus outdoor similarity. Note that curve-based similarity testing of the signals affords the inclusion of the entire gait curve data instead of representative points on the curve used in the parameter-based comparison.

Additionally, we statistically compared the level of dissimilarity by paired comparison of the subject-specific bilateral similarity during treadmill walking versus the similarity between treadmill and outdoor walking. Although natural spatiotemporal asymmetry between left and right sides exists [46], bilateral similarity during treadmill walking was used as a realistic level of similarity, i.e., one of the highest levels of similarity in gait that exists naturally. This level of treadmill bilateral similarity scoring was used as each subject’s baseline and contrasted against the treadmill to outdoor similarity scoring. As each subject’s data was paired with their own control, any potential influence of subject height, leg length, or IMU mounting location on the IMU data would be consistent for each subject’s paired data and, thus, not affect the similarity scores.

Finally, the nonlinear Poincare analysis was deployed on all 12 signals to offer a more descriptive way of assessing the similarity between treadmill and outdoor walking by capturing the changes in the variability when moved from treadmill to outdoor walking at different speeds. The Poincare graph of the two most similar and the two most dissimilar signals of the ranked order analysis was depicted to visualize these variabilities. Secondary outcome measures of peak, range, valley, gait cycle duration, and swing/stance ratio were calculated using the same datasets for comparison of treadmill walking to outdoor walking (Figure 5).

##### Distance Similarity

To measure overall similarity, all data from each participant were aggregated by signal. For these 12 signals, treadmill versus outdoor distance-based similarity measures were calculated (cosine and Euclidean distance). These measures were then ranked for all 12 signals for each of the three walking speeds. 

Cosine distance is defined as the dot product (⋅) of two vectors (v⋅w), i.e., treadmill and outdoor kinematic signals, divided by the product of their magnitudes (|v| and |w|).
(1)$$\mathrm{Cosine}\;\mathrm{distance}\;=\;\frac{\left(v\cdot w\right)}{\left|v|\times |w\right|}=(\hat{v}\cdot \hat{w})$$

Euclidean distance is the distance between the normalized vectors $\hat{v}$ and $\hat{w}$. The square root of the sum of the squares, i.e., norm (${∥\;\cdot \;∥}_{2})$, of the distances in each dimension of the unit vectors.
(2)$$\mathrm{Euclidean}\;\mathrm{distance}\;=\;{∥\hat{w}\;-\;\hat{v}∥}_{2}$$

Cosine has an ideal similarity at 1, and Euclidean distance has an ideal similarity at 0.

A novel process is proposed to compare the dissimilarity between the treadmill and outdoor signals statistically using signal-specific kinematics. Healthy participants’ left and right leg kinematics during indoor treadmill walking should be nearly symmetrical and thus will have the highest level of similarity measure realistically possible for that particular individual [46]. Therefore, this current study defined the similarity of treadmill walking of left and right leg kinematics using Euclidean distance (Equation (2)) for each signal as the signal-specific control value. Next, the similarity for treadmill and outdoor walking for each of the 12 kinematics, i.e., the intervention values, were measured; these were defined as the intervention values. The control value was then statistically compared to the paired intervention values. If significantly different, the Z-score was used to indicate the level of dissimilarities. In summary, we measured the Z-score and *p*-value of the differences between baseline similarity (bilateral symmetry on the treadmill compared to the similarity of the treadmill and outdoor kinematic signals). We termed this analysis Bilateral Symmetry Dissimilarity Testing (BSDT), where a signal-specific control (similarity between treadmill left and right sides) is compared against a signal-specific intervention (similarity between treadmill and outdoor walking signals), as shown in Figure 6.

##### Poincare Similarity

Poincare produces plots of consecutive data points that can be used to quantify measures of short- and long-term variability in a signal [37].

The Poincare analysis is performed using a graphically modified version of the pyHRV library. The standard approach for graphing self-similarity in this recurrence plot was taken by plotting points (xt, xt + 1), (xt + 1, xt + 2), (xt + 2, xt + 3)... An ellipse is then fitted to the resultant scatter plot. The standard deviation along the minor axis of this ellipse (SD1) is a measure of the width of the plot or the *short-term variability* of the temporal data. SD1 is calculated with the standard deviation of successive differences as follows:(3)$$SD1=\sqrt{0.5SDSD^{2}}$$
where *SDSD* is the standard deviation of successive differences (time domain parameters). 

The standard deviation along the major axis of the ellipse (SD2) is a measure of the variance of the total range of values or the *long-term variability* of the temporal data:(4)$$SD2=\sqrt{2SDNN^{2}-0.5SDSD^{2}}$$
where *SDNN* is the standard deviation of the values of the time series. Note that the larger SD1 and SD2, the higher the variability.

#### 2.4.3. Spatiotemporal Parameters

Spatiotemporal parameters (STP) measures were either signal-specific or gait-specific (see Figure 7). This parameter-based testing includes representative points or features of the curve. For each kinematic signal, the signal-specific parameters of peak height (maxima), valley depth (minima), and range (peak-to-valley distance) were extracted for each kinematic signal and then averaged over their respective trial. Note that the selected parameters, i.e., the extremums, of the kinematic signals used in our analysis are shown to be associated with the start and end of the swing/stance phase of gait (heal-strike and toe-off gait events) [10,47,48,49]. For comparison, we used the same features for all of the signals [47,48,49]. Because gait cycle duration and swing/stance time ratio (S/S) are non-signal-specific, they can be derived from one representative signal alone. These gait-specific parameters were derived using ML shank AV. The mediolateral angular velocity of lower limbs, especially that of the shank, is used in various studies for gait segmentation and event detection [50], specifically for measuring gait cycle duration [10,45] and S/S [23,51]. During ambulation, locomotion occurs via rotation around lower limb joints, i.e., around the medial–lateral axis. Thus, angular velocity contains prominent information about participants’ gait and is a reliable kinematic signal for gait analysis. Accordingly, via visual inspection of our dataset, we confirmed the ML shank AV signal as having a relatively low noise ratio across the sensors tested.

The gait-specific parameters, gait cycle duration, and S/S were measured on the ML shank AV, resulting in two values. The three signal-specific parameters, peak height, valley depth, and range, were measured on 12 signals (3D angular velocity and acceleration of shank and thigh, left and right signals combined), resulting in 36 signal-specific values.

### 2.5. Statistical Analysis

Each of the 30 treadmills and 30 outdoor trials was analyzed for the mean and standard deviation for each sensor signal. To compare the treadmill and outdoor spatiotemporal parameters, all AP, ML, and SI data streams were first evaluated with parametric testing using the Shapiro–Wilk test. A paired *t*-test was then used when both datasets being compared were parametrically distributed. In contrast, a Wilcoxon Signed-Ranks test was used if at least one of the compared datasets was nonparametric. The distribution and significance test threshold were defined as less than or equal to 0.05.

Signal similarity measurements were analyzed using comparative and statistical methods. For comparison, the aggregated data from all subjects were ranked by signal, measuring indoor versus outdoor similarity. For the statistical study, individual data from each subject were analyzed as a paired comparison of a bilateral similarity versus treadmill to outdoor similarity.

Similarity measures between indoor treadmill and outdoor walking were statistically compared for all three speeds using paired comparison of each participant’s control group data versus that participant’s intervention group data. The Z-score was also calculated for each signal to provide the level of dissimilarity comparable across all the kinematic signals.

## 3. Results

### 3.1. Walking Speed

Our participants showed an average self-selected walking speed of 1.10 m/s, consistent with previous work that measured self-selected walking speed between 0.75 and 1.67 m/s (Figure 8) [12,16,34]. With fast walking on the treadmill 20% above and slow walking 20% below each participant’s self-selected walking speed, we showed a range from 0.73 m/s (lowest slow walking speed) to 1.74 (highest fast walking speed). Thus, the range of walking speed in this study is essentially at or above a 0.80 m/s threshold, where accuracy drops for wearable sensors on the ankle [2]. The measurement of walking cadence allowed for comparison of indoor and outdoor speed changes between the three speeds. The average cadence for both indoors and outdoors did confirm differences in the walking speeds in the order of 10% magnitude (Treadmill slow 49.01, self-selected 54.08, and fast 56.84; outdoors slow 53.69, self-selected 58.67, and fast 61.54). Thus, we infer the ability to measure gait in both walking conditions at different speeds. 

### 3.2. Signal Similarity Measures

When the aggregate data were ranked by cosine similarity between treadmill and outdoor walking, we observed relative similarity across all 12 combined data streams (cosine similarity score, range 0.9184–0.9996) (Figure 9). This relative similarity was consistent across the tested speeds: the self-selected, slow, and fast. For cosine similarity, there is no defined cutoff for qualifying the different levels of similarity. In our data, however, we detected three distinct bands: the High band was composed of the same five signals across all three speeds, and the Low band was composed of the same three signals across all three speeds. These bands were notable in that while signals showed some intraband variability, there was no interband variability; signals stayed in their respective bands regardless of speed. The High band included SI shank Acc, ML shank AV, SI thigh Acc, ML thigh AV, and AP shank AV (from highest to lowest within the High band, all with a similarity score > 0.9955), all of which moved rank within the High band (intraband variability) (across the three speeds) but did not move into the Middle band at any time (no interband variability). Furthermore, all five High band signals had similarity scores >0.990 at all three speeds. The Low band included SI thigh AV, ML shank Acc, and ML thigh Acc (all with similarity score < 0.973), and these three signals also did not move out of the Low band at any speed, showing no interband variability. The remaining signals compose the Middle band, as shown in Figure 9. Again, although these four Middle band signals shifted rank across the three speeds, demonstrating intraband variability, they did not shift into either the High or the Low band during any speed. 

For Euclidean distance similarity, all aggregated signals remained in the same rank as cosine across all speeds—showing a strong correlation between these two similarity measures across all signals and all speeds. As the ideal cosine similarity is 1 and the ideal Euclidean distance similarity is 0, the perfect correlation of the two signals would be a correlation of −1. In such a case, the cosine would be increasing toward 1 while the Euclidean distance would be decreasing to 0. In this study, the correlation between cosine and Euclidean distance was highly correlated, with a very strong correlation of −0.96 for self-selected walking, −0.96 for slow walking, and −0.97 for fast walking speed.

In the bilateral symmetry dissimilarity testing, individual participant signal data were analyzed and reported in the same banding as delineated by the aggregate signal data. Signals SI shank Acc and ML shank AV showed high similarity between treadmill and outdoor walking across all speeds in the comparative aggregate rank (by ranking in the High band). In addition, these same two signals of SI shank Acc and ML shank AV showed high similarity across all speeds in the bilateral symmetry dissimilarity testing by showing no significant difference across all speeds, as shown in Figure 10. These were the only two signals showing both findings of having high similarity in aggregate testing and similarity in the individual bilateral symmetry dissimilarity testing. In contrast, AP thigh Acc showed similarity across all speeds in the individual data analysis (Figure 10) but did not show high similarity in the aggregate data ranking in the Middle band in the banding in the aggregated data (Figure 9). Therefore, AP thigh Acc was not a reliable kinematic signal across the walking conditions. Other than AP thigh Acc, the remaining signals showed statistically significant dissimilarity between treadmill versus outdoors. The dissimilarity in paired comparison was different across speeds. Self-selected speed identified significant dissimilarity in 8 of the 12 signals, whereas slow speed identified dissimilarity in 7 of the 12 signals, and fast walking speed identified 9 of the 12 as being significantly different.

### 3.3. Poincare Analysis

The ratio of short-term variability (SD1) to long-term variability (SD2) indicators in the ML shank AV (a high-band signal in the ranking) was substantially smaller than the ML shank Acc (a low-band signal with the lowest signal in the ranking) in both treadmill and outdoor walking conditions (see Figure 11). A lower SD1 to SD2 ratio indicates less variability across treadmill and outdoor conditions, which can be defined as having greater robustness across walking conditions. The short-term variability (SD1) and long-term variability (SD2) of ML shank AV demonstrated greater robustness when compared with the ML shank Acc. As can be seen in the High band of Figure 12, only SI shank Acc, one of the most robust signals identified by BSDT, demonstrated no statistically significant differences in long-term (SD2) and short-term (SD1) variability across walking speeds. Two other signals, AP thigh AV from the Middle band and ML thigh Acc from the Low band, also showed no significant differences in SD1 or SD2 across walking speeds. The other BSDT identified robust signal, ML shank AV, demonstrated no statistical differences in SD1 and SD2 at the self-selected and slow walking speeds.

### 3.4. Spatiotemporal Parameters

Spatiotemporal parameters that were measured included both signal-specific parameters, peak, range, and valley depth, and gait-specific parameters, gait cycle duration, and swing/stance time ratio across all three speeds (Figure 13). The signal-specific data was measured from all 12 signals and grouped by the banding, whereas the gait-specific data was measured on ML shank AV.

The statistical testing of the signal-specific parameters showed that all of the signals in the High and the Low band demonstrated a statistically significant difference between treadmill and outdoor walking in at least one of the test speeds (grey boxes). Middle-band signal SI shank AV and Low-band signal SI thigh AV, however, were not significantly different at any of the test speeds across all of the parameters (peak, valley, and range). Slow walking speed showed the lowest level of dissimilarity between parameters compared to the other two walking speeds. The ML thigh AV of the High band was the only signal that showed robustness, measured by non-significant dissimilarity, only at self-selected and slow speeds. Fast walking speed was the most variable, demonstrating significant differences in 10 out of the 12 signals, followed by self-selected speed, eight signals, and slow speed, with only four kinetic signals demonstrating statistically significant differences in any of the parameters. 

For gait-specific parameters, treadmill and outdoor walking gait cycle durations were significantly different regardless of walking speed. The stance/swing ratio (S/S), however, only showed significant differences between treadmill and outdoor walking for slow walking speed.

## 4. Discussion

Defining the optimal kinematic signals, axes, and location(s) for IMU-based gait assessment is challenging. This study aimed to identify these three factors in raw kinematic data signals for use in real-time machine learning. We assessed the differences between treadmill versus outdoor overground walking at three different speeds to identify the optimal kinematic signals for IMU-based gait monitoring systems. A standardized IMU orientation was implemented in which the IMUs were placed in the frontal plane where z = AP axis, y = ML axis, and x = SI axis. Statistically, we identified which raw kinematic signals showed the highest similarity, i.e., reliability when changing walking conditions and speed. Based on our analysis, ML shank AV and SI shank Acc showed the highest similarity banding and the least dissimilarity in paired comparison tests of similarity for treadmill versus outdoor walking across the three walking speeds. Based on these criteria, the above-mentioned kinematic signals were the most robust for gait monitoring applications across the varied walking conditions for our defined sensor placement. When using the aggregated data of all participants, the cosine similarity measure shows distinct high/middle/low banding with ML shank AV and SI Shank Acc, showing amongst the highest similarity scores for treadmill versus outdoor walking. For statistical analysis, a novel test assessing dissimilarity, bilateral symmetry dissimilarity testing (BSDT), showed that the outdoor ML shank AV and SI shank Acc, the aforementioned signals with the highest similarity between treadmill and outdoor walking, were not statistically dissimilar to their overground versions. Among these two signals, SI shank Acc demonstrated the highest level of reliability, being the least variable signal in both BSDT and Poincare analysis while demonstrating robustness in spatiotemporal analysis at self-selected and fast walking speeds.

### 4.1. Identifying the Optimal Kinematic Signals for IMU-Based Gait Monitoring Models

Past work on IMU-based gait monitoring systems included multiple processed kinematic signals from multiple sensors and, thus, came at a high computational cost. In a minimalistic approach, we sought to identify the optimal raw kinematic signals for gait monitoring systems to lower computational costs and increase adoption. To identify these signals, different methods were proposed. Sherifi, Renani, et al. utilized deep learning to predict spatiotemporal gait parameters in patients with osteoarthritis after total knee arthroplasty [52] and, similar to our findings, demonstrated the shank as the location resulting in the highest accuracy in predicting temporal parameters when compared with the foot, thigh, and pelvis locations and their 15 different combinations. Separately, real-time gait phase detection with ML shank AV was implemented by Wang et al. for gait retraining in order to reduce knee adduction moment (KAM) in osteoarthritis [50], where KAM is an important marker associated with disease progression. Using two different neural network models, their gait phase detection accuracy reached 95% [53]. Asuncion et al. [54] used two IMUs in the frontal plane on the thighs to emulate a smartphone carried in a pocket to collect three axes of data from the accelerometer, gyroscope, and magnetometer sensors and then used sensor fusion to provide orientation-independent IMU data for roll, pitch, and yaw for machine learning with a convolutional neural network (CNN). With sensor fusion, compared to individual sensors, they increased the overall accuracy of biometric gait identification by an average of 3–5%, resulting in accuracies between 96.70% and 98.42%. This approach, however, comes with a high computational cost because the learning duration of the CNN increased over three times when using the fused kinematic signals compared to the single kinematic signal method [54]. Reducing the number of signals used in their algorithm to those we identified as robust signals, such as ML shank AV and SI shank Acc, one may decrease the computational burden of their algorithm. While the potential exists for decreasing computational burden, reducing the number of inputs to an AI model may result in poor performance or a complex structure. Here, however, we can find optimal kinematic inputs in a more systematic manner using the comparisons provided here.

### 4.2. ML Shank AV and SI Shank Acc

Our work shows that the ML shank AV and the SI shank Acc signals have the highest level of robustness when walking speed and walking condition change. Previous research has shown that combined gyroscope and accelerometer data gives higher accuracy in gait evaluation than accelerometer data alone [20]. Shank angular velocity and acceleration were successfully utilized for the prediction of freezing of gait in patients with Parkinson’s disease [55], measuring spatiotemporal parameters in typical individuals, stepwise trajectory estimation in young adults [56], monitoring gait normalcy in adults [57], gait event detection in healthy adults and elderly [27,58], real-time gait event detection (swing and stance period initiation) in healthy adults and individuals with spinal cord injury [59,60], and gait phase detection in healthy adults [61] and children with cerebral palsy [9]. Some of these studies used the ML axis for AV [55,59], whereas others used all three axes [56,57]. To evaluate the robustness of these two signals, we methodically assessed their similarity in the face of varying walking conditions. 

### 4.3. Similarity Measures

There is currently no universally accepted gait similarity measure. One early technique of continuous gait pattern analysis (which predates machine learning) was the Gait Deviation Index (GDI) introduced by Schwartz and Rozumalski [38]. GDI was based on singular value decomposition, used in image recognition applications, and compared gait to the gait of control subjects, e.g., comparing the gait of children with CP to those of typically developing children (control subjects). Similarity indices such as GDI can be used to assess continuous gait data across surfaces (as tested in this study), assess participant gait changes, or evaluate for gait pathology [38]. Additional types of continuous data analysis for similarity include the Coefficient of Multiple Determinations (CMD), Statistical Parameter Mapping (SPM), and the Linear Fit Method (LFM) [42,44,62]. Using two gait datasets, 15 healthy and 34 cerebrovascular, of accident patients, Iosa et al. validated a new method of waveform similarity testing, LFM, and showed 94.1% sensitivity and 93.3% specificity when used to compare the two datasets when plotted one versus the other [62]. While the Iosa study advances curve testing and shows promise, it relies on joint kinematic measures, which require more extensive processing steps when compared with using the minimally processed raw kinematic signals. In a separate recent analysis comparing different similarity indices that assess the similarity of curve patterns across the whole gait cycle, Di Marco et al. evaluated RMSD (root mean square deviation), MAV (mean absolute variability, CMC (coefficient of multiple correlation), and LFM (linear fit method) [42]. They concluded that each of these four similarity indices provided partial, but not complete, meaningful comparison (due to differing sources of gait variation) and that each index also came with inherent limitations on the ideal conditions under which it may be valid [42]. Thus, no specific similarity index or protocol is accepted for universal use in the current literature. Building upon cosine and Euclidean distance as similarity measures, we recommend a novel test to assess similarity scores for dissimilarity when comparing a subject’s data across walking conditions. 

### 4.4. Bilateral Symmetry Dissimilarity Testing (BSDT)

In our bilateral symmetry dissimilarity testing, our control condition consisted of left-to-right similarity during treadmill walking as this side-to-side similarity has shown less variability in comparison to treadmill versus overground similarity [46] and can be considered as the subject-specific highest level of symmetry realistically possible for healthy individuals. This concept of utilizing side-to-side symmetry termed bilateral symmetry in this study, was explored by Hill et al. while establishing a healthy control baseline [63]. Hill’s Symmetry Score measured the degree of symmetry between the legs as a baseline in indoor overground walking. Cabral et al. also measured symmetry, identifying a high level of similarity between the left and right legs in typical gait when walking in a controlled environment such as on a treadmill [46]. In the BSDT method, the control dataset is recalibrated for each individual each time they walk on a treadmill to establish an ideal, subject-specific similarity baseline or control. This ideal similarity baseline can then be used for comparison to subsequent intervention, which in this study was outdoor walking. The treadmill symmetry similarity score, as a control condition (both signal-specific and subject-specific control), was statistically compared against the treadmill versus outdoors similarity score. By comparing the bilateral symmetry similarity score, the more robust signals were identified.

### 4.5. Signals: ML Shank AV and SI Shank Acc

BSDT showed no significant difference between our walking conditions in these two signals; however, some significant differences were observed in the less robust signals. Poincare analysis also showed fewer significant differences in the robust signals within the High band when compared to the less robust signals. The STP analysis, however, was less discerning across the signals and did not show a pattern. The decreased variability demonstrated in our Poincare and BSDT analysis, when compared with our parameter-based spatiotemporal analysis, might indicate the strength of curve-based analysis over the parameter-based analysis. Curve-based similarity testing, such as our proposed method BSDT, consists of continuous gait-pattern testing inclusive of the entire continuum of the biomechanical data in the gait curve as opposed to the parameter-based analysis, which utilizes representative discreet features of the kinematic signals. Thus, curve-based analysis might be more versatile in capturing the real differences in the comparing signals and, therefore, is an evolving field within gait analysis. In prior literature, gait comparisons between various conditions are mostly based upon discrete spatiotemporal parameters; this type of parameter-based analysis demonstrated significant between gaits in various walking conditions [34,37], as seen in Figure 13. With any discrete data analysis, however, emphasis on a few specific features or data points may leave out potentially meaningful movement patterns outside those discrete points [59,60]. Such omissions may explain why our STP parameter-based analysis showed more variability in Figure 13 when compared with our curve-based comparison, BSDT, and Poincare, as shown in Figure 10 and Figure 12, respectively. 

### 4.6. Spatiotemporal Differences in Gait

Some previous works noted that the treadmill gait was qualitatively and quantitatively similar to overground walking [45]. Other work showed significant differences between the treadmill and overground gait [. A systematic review noted that 4 of 9 studies showed a difference between treadmill and overground gait [21]. Previous work showed that walking speed affects gait-specific spatiotemporal parameters [13,16,34]. We demonstrated some significant spatiotemporal differences between treadmill and outdoor walking during self-selected and fast walking. However, fewer differences between the treadmill and outdoor walking were detected in slow walking. 

### 4.7. Limitations

#### 4.7.1. Sample Size Limitations

Our participant sample was composed of 30 healthy adolescents, obtained via convenience sampling, and thus limits the generalizability of our results beyond this population. In contrast to past studies in adolescents, which compare either self-selected walking speed treadmill protocols or outdoor walking, our study sample reflects a relatively large sample size. As with all raw data gait research, inter-participant variability and gait perturbations are significant factors that may limit adequate analysis due to limited usable data [2,12]. IMUs generally allow longer periods of gait assessment [22], and we conducted longer gait assessments under both walking conditions, treadmill, and outdoors, as evidenced in the large volume of data collected: 810 min in total walking time by 30 subjects with over 13,200 gait cycles per subject. 

#### 4.7.2. Subjectivity of IMU Studies as a Limitation

Traditional lab gait analysis is well standardized and validated at this point, and the resulting data are comparable across labs regardless of the equipment [64]. The resulting data is comparable across labs regardless of the equipment. In contrast, IMU usage is relatively immature; therefore, IMU study protocols and placement locations are not yet standardized. Systematic reviews report various techniques, making it hard to compare across studies or even participant populations [2,20]. That has led to incongruencies in the data across studies and also encouraged benchmarking of systems against each other [13,65] as there are no consensus guidelines yet for IMU application or reporting [2,20]. Thus, any currently published IMU literature may be constrained by experimental conditions, influenced by the chosen equipment, and/or limited by the data processing procedure for that study. We believe this study minimized this risk by looking at curve-based similarity measures as the prime outcomes measure rather than just the parameter-based comparison (the measured spatiotemporal parameters) from the IMUs alone.

#### 4.7.3. BSDT Limitation

By leveraging widely accepted side-to-side similarity in typical gait, our proposed dissimilarity testing, BSDT, can simply provide subject and signal-specific control values that indicate the magnitude of similarity in different conditions for any curve-based similarity measure. The subject-specificity holds only if the individual has a typical symmetrical gait. In the case of pathological gait, however, one may choose to use signal-specific bilateral symmetry from a normative dataset of typical individuals to conduct a pairwise statistical comparison. 

### 4.8. Future Work

A recent systematic review showed ample published work on young adults comparing treadmill walking to indoor overground walking [21]. However, there is a paucity of work comparing treadmill walking and outdoor walking on typical developing adolescents with IMUs. In this study, a total of 30 adolescent participants were studied walking for a total of 27 min at self-selected, slow, and fast speeds. Twenty-four data streams (four sensors, three axes, and two kinematics) were studied across a total of 317,000 recorded gait cycles. Thus, the dataset resulting from this study can be utilized as a basis for a larger normative database for these typically developing adolescents using IMUs. Additionally, the High-band signals identified in this study included SI shank Acc, ML shank AV, SI thigh Acc, ML thigh AV, and AP shank AV, and it would be productive to confirm and validate these High-band signals with other normative datasets. Future work will include using our analysis techniques with gait data from patients with neuromuscular disease, especially children with cerebral palsy, and evaluating whether the similarity findings of this study are comparable. Ultimately, because the study provides insight into which similarity measures should be considered for machine learning applications, future work could include adapting a machine learning model and comparing outcomes of these identified similarity measures.

Similarity measure testing is not new and dates to square error clustering in the 1960s. However, it is now being revisited as a more relevant topic in machine learning, image processing, and pattern recognition. Due to the high dimensional nature of gait, however, more deliberate and considerate applications of similarity are needed to study gait. It is possible that our use of well-defined statistical similarity measures may be shown to incompletely quantify differences between high dimensional data. As mentioned above, other similarity measures were used for gait data, such as the Coefficient of Multiple Determinations, Statistical Parameter Mapping, and the Linear Fit Method [42,62], and some have introduced new gait-specific similarity measures [44] or gait indices [38]. This myriad of similarity measures, indicating a current lack of standardization of similarity measures, certainly shows that ongoing analysis of larger normative datasets will afford a further understanding of similarity in gait assessment.

## 5. Conclusions

Together, gait data collection, gait analysis, and machine learning enable gait monitoring systems for diagnostics, therapeutics, and assistive technologies. While many previous works have examined parameter-based analysis such as spatiotemporal comparison to quantify gait similarity, we used curve-based similarity analysis to identify reliable kinematic signals when participants change walking conditions and speed. We identified that IMUs aligned in the frontal plane and positioned on the shank were more ideal than IMUs aligned in the frontal plane but located on the thigh. Medial–lateral angular velocity (ML shank AV) and superior–inferior acceleration (SI shank Acc) were the most robust kinematic signals when participants moved from treadmill to outdoor overground walking and at three different walking speeds. These robust raw IMU signals can guide future machine learning-based models for monitoring gait in the face of varying walking conditions while optimizing the processing burden and, thereby, the adaptability of such systems.

## Figures and Tables

**Figure 1 sensors-23-08275-f001:**
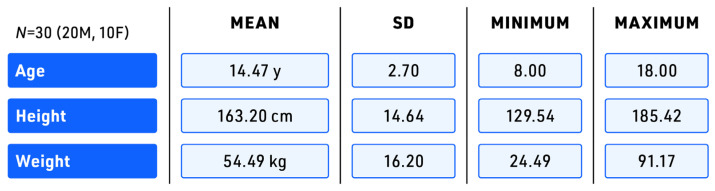
Participant demographics. *N*—number, M—male, F—female, SD—standard deviation, y—years, cm—centimeters, kg—kilograms.

**Figure 2 sensors-23-08275-f002:**
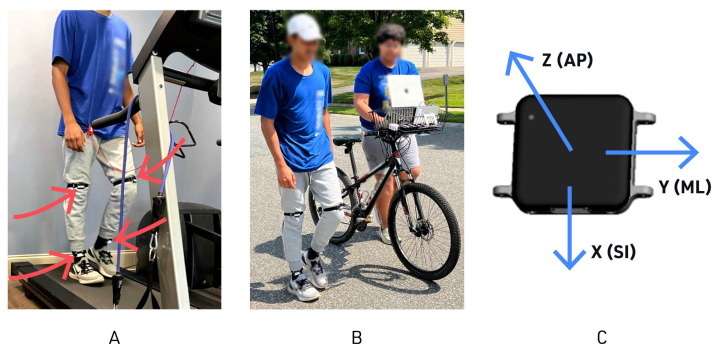
A sample participant wearing mounted IMUs while (**A**) treadmill walking and (**B**) outdoor walking (with customized holder for wireless data collection laptop); (**C**) axis orientation for the inertial measurement units (IMUs): Blue arrows indicate the following plains: Z axis = AP—anterior-posterior, Y axis = ML—medial-lateral, X axis = SI—superior-inferior (Source: APDM Wearable Technologies (v1.0, Portland, OR, USA)).

**Figure 3 sensors-23-08275-f003:**
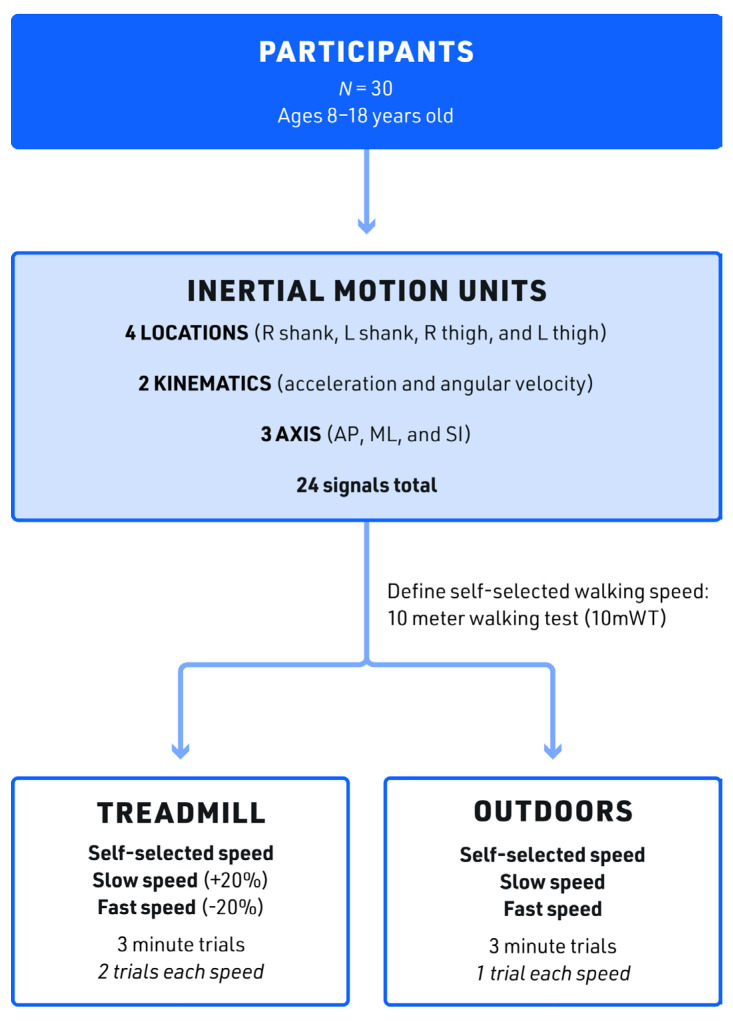
Data Collection: The methodology used in the study involved 30 participants mounted with IMUs at four anatomic locations, recording two kinematic signals along three different axes; walking was measured on the treadmill and outdoors at three different speeds for 3 min for each trial. *N*—number, R—right, L—left, AP—anterior–posterior, ML—medial–lateral, SI—superior–inferior.

**Figure 4 sensors-23-08275-f004:**
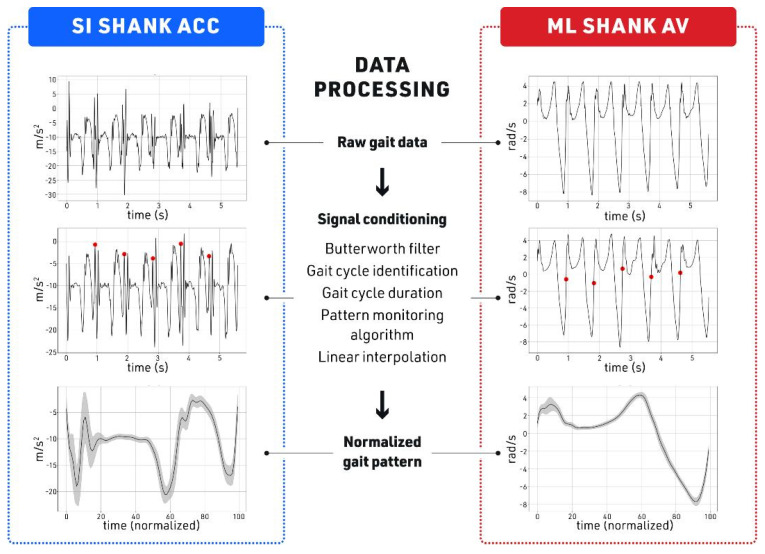
Signal conditioning and data processing. Acceleration in meters per second square (m/s^2^), angular velocity in radian per second (rad/s). Red dots indicate the start of each gait cycle.

**Figure 5 sensors-23-08275-f005:**
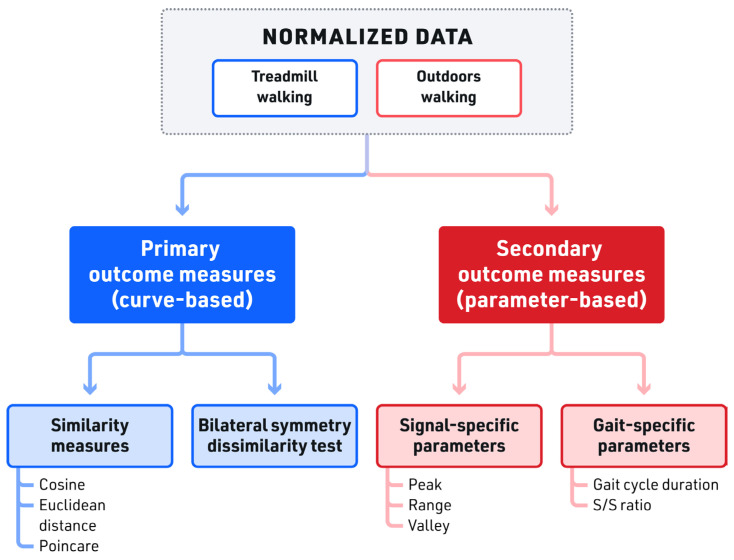
Flowchart of outcome variables. Primary outcomes measures—similarity measures and bilateral symmetry dissimilarity test were calculated in addition to secondary outcome measures of signal-specific and gait-specific parameters.

**Figure 6 sensors-23-08275-f006:**
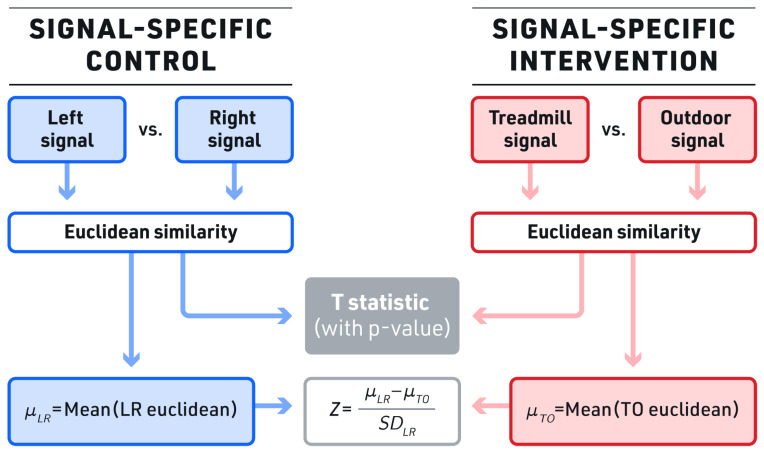
Bilateral symmetry dissimilarity testing. The control group (the similarity of left to right signals) was compared to the intervention group (the similarity of the treadmill to outdoor signals) in the bilateral symmetry dissimilarity testing (BSDT). Both the t value and Z score were calculated to assess for significant differences. LR—left versus right, TO—treadmill vs. outdoors, std—standard deviation, µ—population mean.

**Figure 7 sensors-23-08275-f007:**
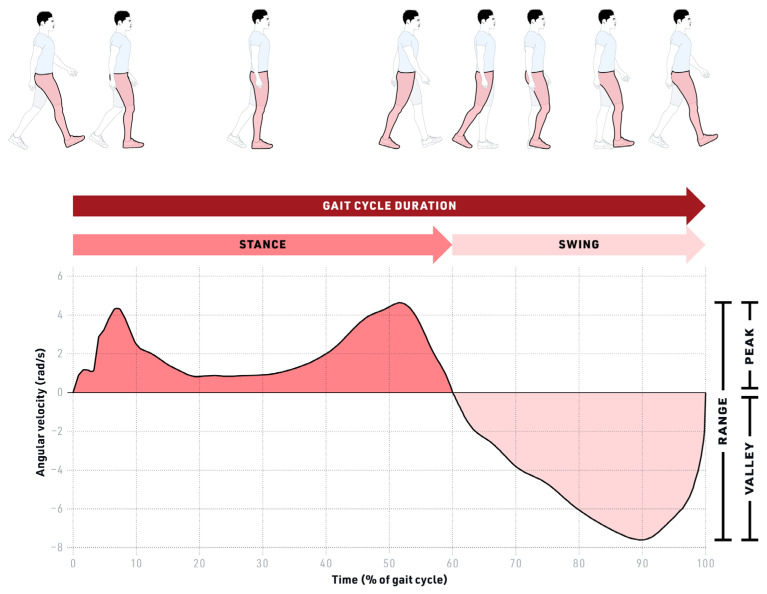
Spatiotemporal (STP) gait parameters. Gait-based parameters: Gait cycle duration and swing/stance time ratio; Signal-based parameters: peak, range, and valley. Rad/s—radian per second, %—percentage.

**Figure 8 sensors-23-08275-f008:**
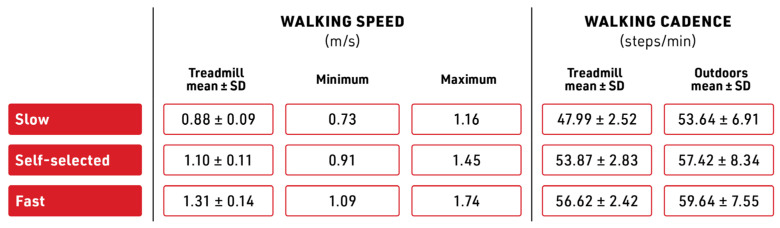
Average walking speeds (slow, self-selected, and fast) for the participant pool. Values within 1 SD around the mean are used to calculate cadence. m/s—meters/second, SD—standard deviation, steps/min—steps per minute.

**Figure 9 sensors-23-08275-f009:**
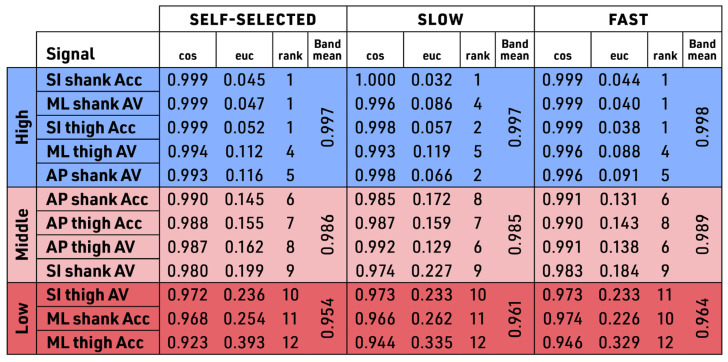
Similarity score banding of group data. Aggregate data for participants showing distinct banding patterns where certain signals remain in respective bands despite changes in walking speeds. The banding pattern is shown with tri-color banding of similarity scores, ranked by cosine and Euclidean distance at a self-selected speed. Bands are shown as High, Middle, or Low based on the aggregate group data. Units: acceleration in (m/s^2^); velocity in (rad/s); Signals: Acc—acceleration, AV—angular velocity, SI—superior–inferior, ML—medial–lateral, AP—anterior–posterior; Speeds: self-selected, slow (−20%), fast (+20%); cos –cosine similarity, euc—euclidian similarity).

**Figure 10 sensors-23-08275-f010:**
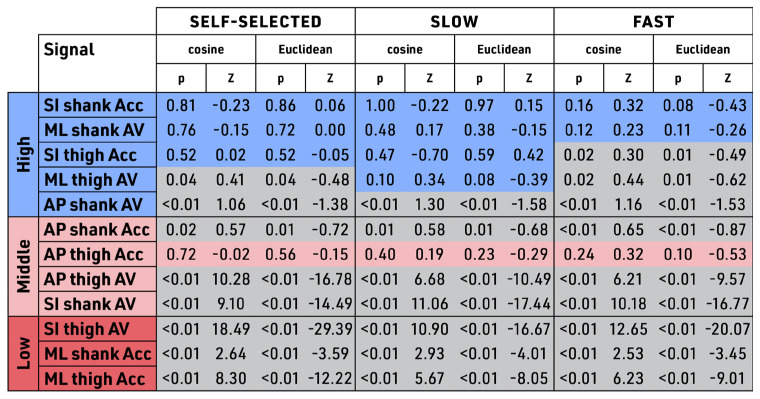
Bilateral symmetry dissimilarity testing (individual data). Tri-color banding similarity score color (non-grey) denotes similarity (signals with no significant difference for cosine and Euclidean distance for control vs. intervention groups). Grey denotes dissimilarity (significantly different cosine and Euclidean distance for control vs. intervention groups). Bands are shown as High, Middle, or Low based on the aggregate group data; Units: acceleration in meter per second square (m/s^2^); velocity in radian per second square (rad/s); Signals: Acc—acceleration, AV—angular velocity, AP—anterior–posterior, ML—medial–lateral, SI—superior–inferior; Speeds: self-selected, slow (−20%), fast (+20%).

**Figure 11 sensors-23-08275-f011:**
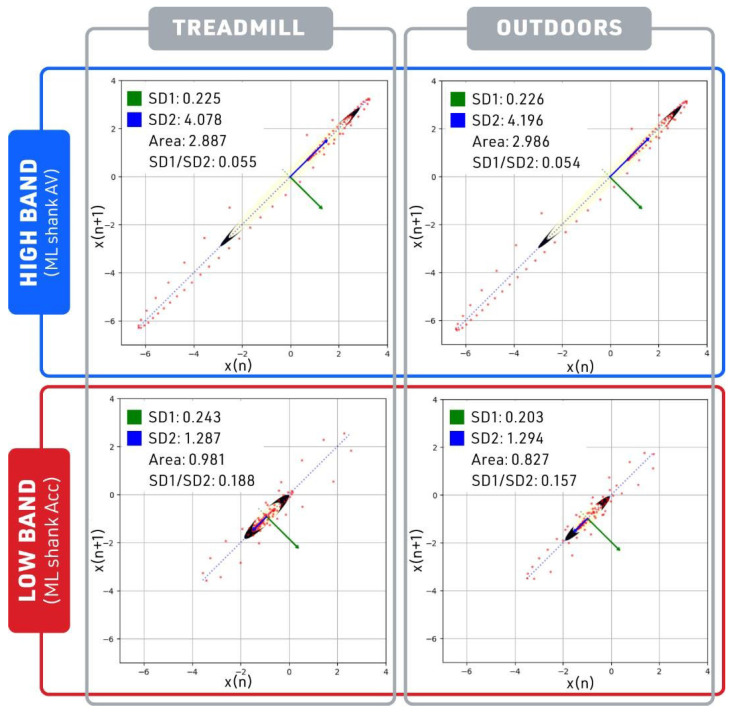
Poincare analysis of self-selected speed. Plot to visualize a High-band signal (Mediolateral shank angular velocity, shown in blue box) and a Low-band signal (mediolateral shank acceleration, shown in red box) from the treadmill and outdoor datasets.

**Figure 12 sensors-23-08275-f012:**
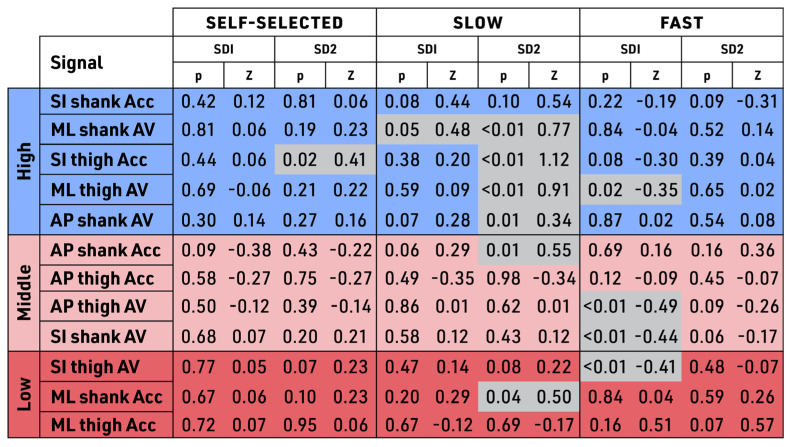
Poincare analysis: pairwise evaluation of similarity of treadmill versus outdoors (by SD1 and SD2). SD1 is short-term variability, SD2 is long-term variability with a significance level of *p* < 0.05, and Z score is shown. Tri-color banding similarity score color (non-grey) denotes similarity (signals with no significant difference for cosine and Euclidean distance for control vs. intervention groups). Grey denotes dissimilarity (significantly different cosine and Euclidean distance for control vs. intervention groups). Bands are shown as High, Middle, or Low based on the aggregate group data; Units: acceleration in meter per second square (m/s^2^); velocity in radian per second square (rad/s); Signals: Acc—acceleration, AV—angular velocity, AP—anterior–posterior, ML—medial–lateral, SI—superior–inferior; Speeds: self-selected, slow (−20%), fast (+20%).

**Figure 13 sensors-23-08275-f013:**
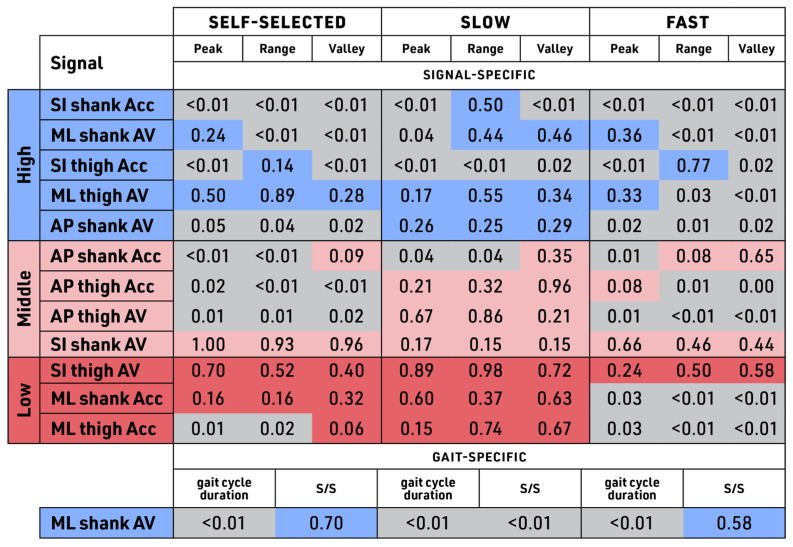
Spatiotemporal parameters (STP by band and across speeds). Comparison of the treadmill to the outdoors for individual data where significance level, *p* < 0.05. Tri-color banding similarity score color (non-grey) denotes similarity (signals with no significant difference for cosine and Euclidean distance for control vs. intervention groups). Grey denotes dissimilarity (significantly different cosine and Euclidean distance for control vs. intervention groups).

## Data Availability

All data analyzed for the study’s findings can be found at https://shorturl.at/ekmZ9, accessed on 1 October 2023.

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
