# Peer review of "Enhancing Wearable Gait Monitoring Systems: Identifying Optimal Kinematic Inputs in Typical Adolescents"

_sensors, 2023, doi:10.3390/s23198275_

Round 1

Reviewer 1 Report

This paper aims to identify robust kinematic signals for gait analysis by employing signal similarity analysis across diverse walking conditions. Although the paper's original idea and the problem it addresses are of interest, the authors' approach to tackle the problem raises some concerns, casting doubt on the veracity of the findings and conclusions drawn in the study. My major concerns regarding the methodology are the following:

1. A foundational premise of the work assumes that kinematic signals from treadmill and outdoor overground walking share analogous characteristics and thus should display congruent data patterns. Yet, existing literature elucidates disparities between treadmill and overground walking. Hence, divergences noted in similarity may be indicative of these intrinsic disparities rather than indicating diminished reliability of the kinematic signals.

2. The study exclusively involves healthy participants, rendering the conclusions not applicable to other populations where the signals identified as robust may not possess the sensitivity to discern variations between healthy and unhealthy groups. As a result, the scope of applicability of your conclusions remain constrained. 

3. The outdoor walking trials failed to control for the participants' walking speed, thereby compromising the comparability of results. Unlike treadmill experiments that imposed a fixed walking speed, overground walking naturally involves fluctuating gait speeds.

4. The inclusion of adolescents spanning ages 8 to 18 introduces potential discrepancies in participant height and leg length, factors that can significantly influence inertial sensor data. The absence of strategies to mitigate these variations undermines the validity of the comparisons.

5. The comparison between right and left sensor data presupposes symmetry in gait patterns. However, symmetry doesn't necessarily translate to similarity in inertial sensor data. To enhance the comparability of left and right sensor data and mitigate sensor data symmetry, a reflection process should be employed on the sensor data.

Moreover, the following aspects require further consideration:

1. The title is too big, please consider shortening it.

2. The acronym IMU typically stands for Inertial Measurement Unit; however, different translations of this acronym are used throughout the text.

3. In lines 211 to 212 it should be make clearer that the x, y, and z axes of the IMUs were aligned with SI, ML and AP axes of the body; when the sensor is not placed in the body (like illustrated in Fig 1 C) the nomenclature of AP, ML, and SI is not rigorous.

4. How did you measure the walking distance in outdoor trials?

5. How were ranks of the kinematic signals established? In other words, what was the methodology employed to convert distance values into their corresponding ranks?

6. Please clarity whether the 4th order low pass filter was consistently applied to all sensor data. Do you think the threshold of 20Hz is appropriate for all walking speeds under consideration? Additionally, could you specify whether the similarity measures were derived from filtered data or the raw data? Furthermore, I would appreciate an elaboration on how this preprocessing might influence the outcomes of the study.

7. The exclusion of "aberrant" gait cycles predicated solely on an average gait cycle duration devoid of considerations for individual walking speeds. However, the average gait cycle duration should vary considerably with gait speed and height, and thus lead to ineffective exclusion of gait cycles. Alternatively, a subject and speed-specific approach could have been applied. Could you further elaborate on this aspect?

8. The nomenclature of control and intervention (in lines 408-410) does not seem appropriate in this context. I would consider it appropriate if the control would refer to two different trials of the same participant under the same conditions. Could you please further elaborate on the choice of these terms?

9. Lines 481 and 482 mention a descriptive analysis that isn't subsequently presented.

10. The application of the Shapiro-Wilk test for assessing normal distribution warrants clarification, including whether this test compared data streams or similarity measures.

11.  Walking cadence presented in lines 512-513 are unitless and not consistent with those presented in Table 2. Moreover, Table 2 states that walking cadence was measured in m/s like gait speed, which seems to be incorrect.

12. In Table 2, self-selected, slow, and fast speeds are not consistent as slow speeds are higher than self-selected and fast speeds. Since walking speeds are presented as an average +- SD, walking cadence should be presented in a similar manner.

13. In lines 528-543 you elaborate on the formulation of three distinct bands. I understood that these bands were conceived to ensure uniform rank inclusion across various walking speeds. However, I'm curious about the underlying logic that guided this partitioning. Could you elucidate the principles that guided the determination of thresholds and the rationale for their selection? Furthermore, it's noteworthy that in the context of fast walking speeds, the ranks 10 and 11 display inconsistency between the middle and low ranks. Given this inconsistency, why wasn't SI thigh AV incorporated in the low rank to foster uniformity?

14. In Fig 7, what gait speed was considered in the plot? Is it aggregating all gait speeds and subjects?

15. In line 670, sensor axes are associated with body planes; however, it's crucial to note that each plane can be defined by two perpendicular axes, rendering this association inaccurate.

16. The assertion made in lines 711-713, suggesting that reducing the number of signals diminishes computational burden, warrants reevaluation. While it is true that a decreased signal count might indeed lighten the computational load, it's important to recognize that opting for fewer signals could potentially necessitate more complex deep learning architectures. Could you please further elaborate on this matter?

17. Throughout the discussion, you consistently emphasize the demonstration of signal robustness amid shifts in walking speed and terrain. Yet, it's essential to acknowledge that transitioning from a treadmill to an outdoor setting doesn't equate to altering the terrain itself, as other substantial differences exist between the two. For this claim to be scientifically sound, a comparison encompassing diverse walking terrains—such as asphalt, soil, sand, and the like—would be necessary.

18. The two signals identified as having more robust properties (ML shank AV and SI shank Acc) are also those expected to have higher signal-to-noise ratio due to the nature of the movement and the sensor's close proximity to the ground. Could signal normalization and more tailored filtering be deemed necessary to effectively address signal disparities and enhance comparability between them?

19. In order to derive overarching conclusions about the robustness of the signals from the extracted parameters, it's essential to ensure a robust parameter extraction process.  Can you ascertain the robustness of your parameter extraction process? 

20. In Table 5, how were the p-values determined? Were they obtained by comparing the data of each subject between treadmill and overground walking conditions?

Author Response

Attached is the document for response to reviewer 1.

Reviewer 2 Report

The paper presents gait assessment using IMU for varying walking conditions for developing adolescent. The paper has merit in biomechanics research and relevant to the journal. However, the authors need to address a few issues.

1) The title is quite long. It is better if the author could revise, make it specific.

2) The problem statement is not clear in Introduction part, that lead to the objectives of this study.

3) The main objectives of the study also not really clear since the author put to much comparisons in the statement. The authors should focus on the final outcomes that they expected from those comparisons.

4) The details of statistical power calculation based on the number of subjects should be provided

5) Since the range of subjects' age is quite large (8-18), how do you generalize the findings? The anatomical size and gait behaviour must be affected during experiment.

Author Response

Attached is the document for response to reviewer 2.

Round 2

Reviewer 1 Report

I thank the authors for answering all my previous comments, which enhanced my understanding of the work. Most of my previous comments have been sufficiently addressed, leading me to believe that the paper is now suitable for publication.

Before acceptance, I would recommend authors to conduct a thorough review of the text to identify and correct all typos.

Author Response

Thank you -- will submit the updated, revised manuscript that has been scanned for typos.